# Differences in Alternative Splicing between Yellow and Black-Seeded Rapeseed

**DOI:** 10.3390/plants9080977

**Published:** 2020-07-31

**Authors:** Ai Lin, Jinqi Ma, Fei Xu, Wen Xu, Huanhuan Jiang, Haoran Zhang, Cunmin Qu, Lijuan Wei, Jiana Li

**Affiliations:** 1Chongqing Rapeseed Engineering Research Center, College of Agronomy and Biotechnology, Southwest University, Chongqing 400715, China; linaiiiiii@email.swu.edu.cn (A.L.); mjq2014@email.swu.edu.cn (J.M.); swuxf9310955@email.swu.edu.cn (F.X.); xw1243524211@email.swu.edu.cn (W.X.); jh8469259@email.swu.edu.cn (H.J); zhr840905528@icloud.com (H.Z.); drqucunmin@swu.edu.cn (C.Q.);; 2Academy of Agricultural Sciences, Southwest University, Chongqing 400715, China

**Keywords:** alternative splicing, RNA-seq, WGCNA, yellow seed, *Brassica napus*

## Abstract

Yellow seed coat color is a desirable characteristic in rapeseed (*Brassica napus*), as it is associated with higher oil content and higher quality of meal. Alternative splicing (AS) is a vital post-transcriptional regulatory process contributing to plant cell differentiation and organ development. To identify novel transcripts and differences at the isoform level that are associated with seed color in *B. napus*, we compared 31 RNA-seq libraries of yellow- and black-seeded *B. napus* at five different developmental stages. AS events in the different samples were highly similar, and intron retention accounted for a large proportion of the observed AS pattern. AS mainly occurred in the early and middle stage of seed development. Weighted gene co-expression network analysis (WGCNA) identified 23 co-expression modules composed of differentially spliced genes, and we picked out two of the modules whose functions were highly associated with seed color. In the two modules, we found candidate DAS (differentially alternative splicing) genes related to the flavonoid pathway, such as *TT8* (*BnaC09g24870D*), *TT5* (*BnaA09g34840D* and *BnaC08g26020D*), *TT12* (*BnaC06g17050D* and *BnaA07g18120D*), *AHA10* (*BnaA08g23220D* and *BnaC08g17280D*), *CHI* (*BnaC09g50050D*), *BAN* (*BnaA03g60670D*) and *DFR* (*BnaC09g17150D*). Gene *BnaC03g23650D*, encoding RNA-binding family protein, was also identified. The splicing of the candidate genes identified in this study might be used to develop stable, yellow-seeded *B. napus*. This study provides insight into the formation of seed coat color in *B. napus*.

## 1. Introduction

Alternative splicing, which results in a single gene coding for a variety of proteins [1], is a crucial mechanism for regulating gene expression. The phenomenon was first observed by Gilbert in 1977 [2], and first characterized in 1981 [3]. During the processing of a gene’s primary transcript (pre-mRNA), particular segments of the gene may be included or excluded from the final messenger RNA (mRNA), resulting in alternative transcripts for the gene. Alternative splicing occurs frequently in many organisms, where it greatly increases the diversity of encoded proteins in the genome [4]. For example, the proportion of intron-containing genes undergoing alternative splicing (AS) is over 92% in human (*Homo sapiens*) [5], 60% in *Arabidopsis thaliana* [6], 52% in soybean (*Glycine max*) [7], 40% in maize (*Zea mays*) [8] and 40% in *Gossypium raimondii* [9]. Five basic modes of AS events are generally recognized: exon skipping (ES), mutually exclusive exons (ME), alternative donor site (AD), alternative acceptor site (AA) and intron retention (IR) [10], although other modes of alternative splicing have been observed. Exon skipping is the most common mechanism of AS in mammals, whereby a particular exon may be included or excluded from a gene’s mRNA under some conditions, or in particular tissues [11]. In plants, however, the most frequent AS event is intron retention [7].

mRNA is spliced by the spliceosome, an RNA–protein complex consisting of small nuclear ribonucleoprotein (snRNP) modules and a variety of auxiliary proteins [12]. Splice sites are determined by cis-acting elements, trans-acting RNA-binding proteins (RBPs) and splicing repressor proteins [13] such as exonic splicing enhancers, intronic splicing silencers and so on [14]. A typical eukaryotic nuclear intron contains consensus sequences that are of vital importance for splicing. Nearly all plant introns have the sequence GU at the 5’ end. Near the 3’ end there is a branch point which is followed by a series of pyrimidines (the poly-pyrimidine tract) and then by AG at the 3’ end. This consensus sequence is known as a “GU-AG boundary”, and introns containing this sequence follow the “GU-AG” rule [15]. This kind of intron is classified as a U2 type intron, the most common type of intron in eukaryotes (the other being the U12 type) [16].

Alternative splicing of mRNA precursors greatly increases the diversity of mRNA and protein isoforms, and the process influences all aspects of organisms. In plants, alternative splicing contributes to cell differentiation [17], organ development [8], acquisition and maintenance of tissue identity [1] and responses to biotic and abiotic stresses such as heat and disease [18]. It was reported that constitutive and inducible alternative splicing of OsWRKY62 and OsWRKY76 in rice (*Oryza sativa*) may help defend against the blast fungus *Magnaporthe oryzae* and the leaf blight bacterium *Xanthomonas oryzae pv oryzae* [19].

*Brassica napus* is one of the most important oil crops in the world [20]. Seed coat color is an important trait in *B. napus*. Yellow-seeded *B. napus* is more valuable than black-seeded *B. napus*, because of its higher oil content and better-quality meal [21]. Yellow-seeded *B. napus* also has other advantages over black-seeded varieties: the seed coat is thinner and the oil is clearer, which not only retains the original flavor of the vegetable oil but also gives it a good commercial appearance. To identify the candidate genes for seed coat color, previous researchers mainly performed QTL mapping analysis [22], and focused on the flavonoid biosynthesis pathway [23]. The mutation of genes in the flavonoid synthesis pathway results in defective pigment and proanthocyanidins synthesis, which might form an abnormal or transparent seed coat. Thus, yellow seeds show the color of the embryo due to the colorless seed coat [23]. However, to our knowledge, there have been no post-transcriptional regulatory studies about the formation mechanism of yellow-seeded *B. napus*.

In this study, we analyzed alternative splicing in yellow- and black-seeded *B. napus*, and identified the DAS genes associated with seed color in *B. napus*. The candidate genes involved the flavonoid pathway, and some genes encoded the SER/ARG-rich (SR) protein and RNA-binding family protein.

## 2. Results

### 2.1. Putative Transcript Assembly

To understand the role of alternative splicing (AS) in black-seeded and yellow-seeded *B. napus*, 31 RNA-seq data of seeds in five developmental stages after flowering (14, 21, 28, 35 and 42 d) were obtained [24] and subjected to quality trimming, followed by transcript assembly and transcript merging using the RNA-seq data-analysis pipeline. We identified 4249, 5164, 5953, 6341 and 6438 new transcripts in yellow-seeded *B. napus* for the five developmental stages, and 4684, 5747, 6427, 6626 and 7000 in black-seeded *B. napus* with the class code “u”, none of which exist in the reference genome (Appendix A). We also obtained the FPKM values for the genes and isoforms. The average FPKM values of the yellow- and black-seeded *B. napus* at the five developmental stages were 18.09, 54.60, 20.19, 16.17, 20.45, 12.67, 37.74, 18.10, 73.10 and 250.19, respectively (Appendix A), whereas the average FPKM values for the new transcripts were 33.64, 593.20, 85.92, 32.46, 22.04, 16.78, 282.08, 44.89, 564.86 and 645.39. Thus, the gene expression levels were higher for the new transcripts.

### 2.2. Analysis of the AS Landscape in Yellow- and Black-Seeded *B. napus*

The AS patterns and landscapes of yellow- and black-seeded *B. napus* were identified using Astalavista-4.0. We detected 21,931, 20,142, 21,237, 18,842 and 19,801 AS events in five developmental stages of yellow-seeded *B. napus*, respectively. We also detected 20,248, 20,366, 23,519, 19,806 and 20,471 AS events in five developmental stages of black-seeded *B. napus*, respectively. By comparing the AS patterns, we found that IR is the most common one, followed by AA, AD and ES (Figure 1A, Appendix A).

The results indicate that the AS events in five developmental stages of yellow- and black-seeded *B. napus* were almost same over the course of seed development, and that IR accounted for a large proportion of the AS events. The AS patterns were highly conserved in all samples.

### 2.3. Differential AS gene

We detected 2733, 2743, 4177, 3655 and 3144 DAS genes in five developmental stages, 14, 21, 28, 35 and 42 d after flowering, respectively (Figure 1B). A total of 830 overlapping DAS genes in all five developmental stages between yellow- and black-seeded samples were identified (Figure 1B, Appendix A). These genes mainly participate in metabolic/catabolic processes, protein ubiquitination, potassium ion import, DNA repair and short-day photoperiodism (Figure 1C). Three DAS genes encoding SER/ARG-rich (SR) protein (*BnaC08g21130D*, *BnaC01g26160D* and *BnaA06g15930D*) were identified, which can bind to some particular RNA sequences and assemble the spliceosome at weak splice sites, and thus play a significant role in the constitutive and alternative splicing of pre-mRNA in rice [25]. We also identified DNA/RNA-binding protein (*BnaC04g41260D*, *BnaAnng15850D*, *BnaC03g02460D* and *BnaC07g01360D*), which has been reported to be involved in circadian clock regulation [26] with an hnRNP-like RNA-binding protein in *A. thaliana* [27]. We also found some transcription factors, such as the basic region/leucine zipper transcription factor 16 (bZIP16).

We also identified 646, 540, 1379, 917 and 748 specific DAS genes in seeds at 14, 21, 28, 35 and 42 d after flowering, respectively. After performing Gene Ontology (GO) analysis of specific DAS genes in five developmental stages of yellow- and black-seeded *B. napus* (Appendix A), we observed that in the early and middle stages of seed development (14, 21 and 28 d), the DAS genes were mainly enriched in mRNA splicing. The specific DAS genes in seeds at 35 d after flowering were mainly involved in monocarboxylic acid metabolic process. While at 42 d after flowering, the specific DAS genes were mainly enriched in localization, organophosphate metabolic process, and catabolic processes of aromatic compounds, heterocycles, cellular nitrogen compounds and organic cyclic compounds. The results indicate that AS mainly occurred in the early and middle stages of seed development. In addition, we found DAS genes involved in flavonoid metabolism, such as *transparent testa* genes (*TT5*: *BnaC02g38340D*, *BnaC08g26020D*, *TT8*: *BnaC09g24870D*, *TT10*: *BnaAnng08030D*, *BnaA06g30430D*, *BnaA09g34840D*, *TT12*: *BnaC06g17050D*, *BnaA07g18120D*, *TT16*: *BnaAnng30140D* and *BnaC09g04950D*) and *seedstick* (*STK*) (*BnaAnng39120D* and *BnaCnng46740D*) (Table 1).

### 2.4. Identification of Modules Associated with Seed Color by Construction of Weighted Gene Co-Expression Network and Correlation between Each Module

To uncover the major candidate genes for seed color, we performed weighted gene co-expression network analysis (WGCNA) with all DAS genes. WGCNA can group similarly expressed genes into modules or networks based on pairwise correlations between genes [28]. Using a soft threshold of 5, the co-expression network was generated and 23 modules were obtained (Figure 2A,B).

Meanwhile, we performed GO analysis to explore module function, with two modules (black and dark grey modules) related to flavonoid metabolic pathways that have been reported to be important for seed color (Appendix A). The function of the other 21 modules is also shown in Appendix A. The genes in the black module are mainly involved in the flavonoid biosynthetic and metabolic process, the proanthocyanidin metabolic process, the fatty acid catabolic process and the maintenance of seed dormancy. The genes in the dark grey module are mainly involved in the anthocyanin-containing compound biosynthetic process, the fatty-acyl-CoA metabolic process, and the fatty acid derivative metabolic process.

Using a weight value of >0.15, we performed a Cytoscape network analysis for the two vital modules to show the relationships among genes in a single module (Figure 3 and Appendix A). The black and dark grey modules contained 171 and 25 genes, respectively, and the most and least highly connected genes are shown in red and purple, respectively.

Among the black and dark grey modules, the candidate DAS genes involved in flavonoid metabolism were identified, such as TT8 (*transparent testa 8*, *BnaC09g24870D*), TT5 (*BnaA09g34840D* and *BnaC08g26020D*), TT12 (*BnaC06g17050D* and *BnaA07g18120D*), AHA10 (autoinhibited H(+)-ATPase isoform 10, *BnaA08g23220D* and *BnaC08g17280D*), CHI (chalcone-flavanone isomerase family protein, *BnaC09g50050D*), BAN (BANYULS, *BnaA03g60670D*) and DFR (dihydroflavonol 4-reductase, *BnaC09g17150D*) (Figure 3 and Table 1). In addition, we found one gene (*BnaC03g23650D*) that encodes RNA-binding family protein. These DAS genes might play a major role in the formation of seed coat color.

## 3. Discussion

In this study, we revealed the AS landscape and identified DAS genes between yellow-seeded and black-seeded oilseed rape. We performed GO analysis on 830 overlapping markedly DAS genes. The major GO terms of these genes include transport, regulation of protein homodimerization activity, establishment of localization, organic substance transport and exocyst assembly.

In order to identify the candidate genes involved in seed color, we used WGCNA to classify the DAS genes into 23 modules. Among the 23 modules, we performed GO analyses and module-sample cluster analysis to choose two modules related with seed coat color formation. It has been reported that pigments and other chemical substances affect the seed coat color during seed coat formation [23]. Flavonoids are a class of low-molecular-weight phenolic compounds, and are representative plant secondary products. In addition, flavonoids play a vital role in sexual reproduction and in protecting plants from UV damage [29]. The chemical diversity of flavonoids can be increased enormously by tailoring reactions that modify scaffolds, including glycosylation, methylation and acylation [30]. Based on the above pathways, we chose modules involved in flavonoid, lignin, phenylpropanoid biosynthetic and metabolic processes, as well as pigmentation and anthocyanin accumulation, all of which correspond to previously reported results [23,31].

Alternative splicing in flavonoid genes or transcription factors (TFs) affect the color of the plant. In peach flowers, the *ANS* gene of white flowers performed intron retention of a spare 129 bp sequence lacking in pink flowers, which leads to the dysfunction of the protein, and finally blocks the accumulation of colored pigments in white petals [32]. In tomato, splicing mutations of transcription-factor-encoding genes which control the loci of anthocyanin pigmentation leads to a loss of function of the wild-type protein. This is why domesticated tomato fruits display a red coloration instead of the purple color in wild species [33]. According to the functional annotation of *Arabidopsis*, homologous genes of 11 DAS genes were related to the seed color: TT8, two TT5 genes, two TT12, two AHA10, CHI, BAN, DFR and *BnaC03g23650D*. TT8 encodes an enzyme that regulates the flavonoid pathway [34] and a TF involved in the regulation of anthocyanin [35]. The expression of TT8 is strongly connected with seed coat differentiation in *Arabidopsis thaliana* [34]. TT5 is involved in the synthesis of chalcones in the flavonoid pathway [36]. TT12, a kind of transporter of proanthocyanidins (PAs) [37], is suggested to regulate the accumulation of proanthocyanidin in the seed [38]. It has been shown that the relative expression of TT12 in brown cotton was higher than in white cotton [37]. AHA10 has been proven to be the early biosynthetic gene involved in flavonoid biosynthesis [23]. Highly expressed AHA10 was involved in the acidification of lemon (*Citrus limon (L.) Burm.*) [39]. CHI encodes the second enzyme of flavonoid biosynthesis; it catalyzes chalcone into corresponding flavones [40]. Research has shown that BAN is closely associated with *Arabidopsis thaliana* seed coat [41]; it codes the central enzyme of proanthocyanidin biosynthesis [35]. DFR is the late biosynthetic gene of flavonoid biosynthesis; it participates in catalytic synthesis of anthocyanin and procyanidins in plants [42]. The DAS genes found in this study might play a major role in the formation of seed coat color in *B. napus*, which needs to be verified in the future.

We also selected some crucial AS genes, such as *STK*, genes encoding SR protein and TF bZIP6. STK, a transcription factor that could facilitate initiation and growth of ovule [43], has also been proven to regulate cell wall properties of the seed coat [44]. SR protein plays vital roles in the constitutive and alternative splicing of pre-mRNA [25]. It has been proven to enhance splicing and alter the splice sites in rice, showing the significance of the SR domain for the enhancement of splicing efficiencies [25], thus regulating the growth and development of plants. Research has shown that bZIP16 promotes seed germination and hypocotyl elongation by repressing the expression of some responsive hormone genes [45]. In this study, we explored candidate genes related to seed color through AS, thereby providing more possibilities to develop stable yellow-seeded *B. napus*.

## 4. Materials and Methods

### 4.1. Downloading of RNA-Seq Data

We downloaded RNA-seq data from the seeds of yellow- (cultivar: No.2127-17) and black-seeded (cultivar: 94,570) *B. napus* in five developmental stages, 14, 21, 28, 35 and 42 d seeds after flowering, with three or four biological replicates: SRR5590338, SRR5590943, SRR5590945 and SRR5591364 (14 d; black-seeded); SRR5590946, SRR5590955 and SRR5590957 (14 d; yellow-seeded); SRR5590959, SRR5590960 and SRR5590972 (21 d; black-seeded); SRR5590973, SRR5590975 and SRR5591012 (21 d; yellow-seeded); SRR5591412, SRR5591464 and SRR5591481 (28 d; black-seeded); SRR5591522, SRR5591524 and SRR5591597 (28 d; yellow-seeded); SRR5591631, SRR5591634 and SRR5591709 (35 d; black-seeded); SRR5591714, SRR5591715 and SRR5591717 (35 d; yellow-seeded); SRR5591718, SRR5591719 and SRR5591720 (42 d; black-seeded); and SRR5591722, SRR5591723 and SRR5591744 (42 d; yellow-seeded) [46].

### 4.2. Analysis of Processed RNA-Seq Data and AS Landscape

The original data were converted using the SRA Toolkit 2.9.0 [47], and quality trimming was performed with Trimmomatic-0.36 [48] (modified parameters: ILLUMINACLIP: TruSeq3-PE.fa:2:30:10, LEADING:3, TRAILING:3, and SLIDINGWINDOW:4:15, MINLEN:50). Then, the processed groups of data were mapped to the *B. napus* reference genome [49] using STAR-2.5.3a [50] (with sjdbOverhang = 150 and limitBAMsortRAM = 5,189,162,595), and Cufflinks-2.2.1 was used to assemble putative transcripts [51]. The subsequent program Cuffcompare could map the different isoforms to the corresponding genes [52], and the last step was to analyze the AS patterns and to visualize the AS landscape using Astalavista-4.0 [53].

### 4.3. Construction of Weighted Gene Co-Expression Networks and Analysis of Overlapping Differentially Spliced Genes

WGCNA, a freely accessible R package for the construction of weighted gene co-expression networks [54], can help identify gene modules of interest from thousands of seed coat genes. WGCNA in R version 3.4.4 was used to perform sample clustering, outlier detection, soft threshold filtering, module identification and module relationship analysis. In our study, the one-step function was used to construct networks and detect consensus modules.

### 4.4. Analysis of Overlapping Differentially Spliced Genes

We used rMATS [55] under a Python console to investigate the differential alternative splicing between yellow- and black-seeded samples. The rMATS method allows analysis of all major types of alternative splicing patterns and uses RNA-Seq reads, mapped to both exons and splice junctions [55]. Fastq format files were uploaded and the STAR index file of the genome was used for the comparative analysis.

### 4.5. Functional Enrichment Analysis of Genes in Every Module

The genes inside the co-expression modules identified by WGCNA have high connectivity and similar function. We performed the R package topGO to perform the GO analysis [56]. We filtered the hub genes in each module according to the intra-modular connectivity and further understood their function.

## 5. Conclusions

In this study, we identified candidate alternative splicing genes between yellow-seeded and black-seeded oilseed rape during the post-transcriptional regulatory process. The function of the candidate genes needs to be verified by molecular biology experiments in the future. This study will help us understand the formation mechanism of yellow-seeded *B. napus*.

## Figures and Tables

**Figure 1 plants-09-00977-f001:**
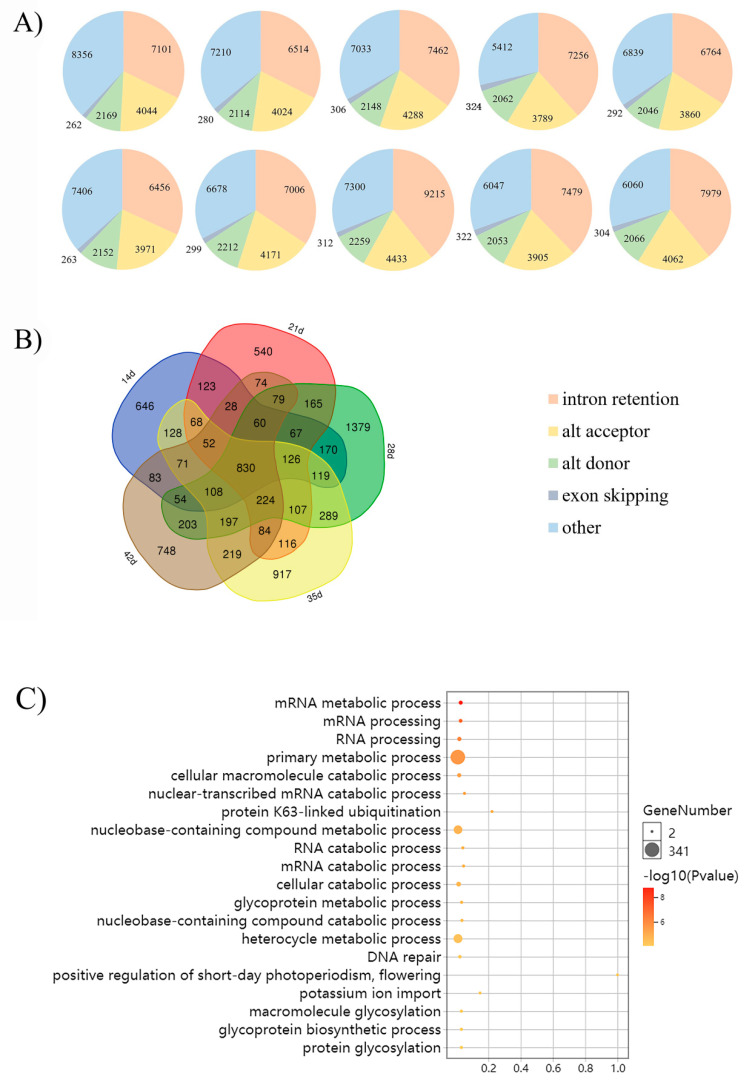
The alternative splicing (AS) landscape of yellow- and black-seeded *B. napus* in five developmental stages of seed development (14, 21, 28, 35 and 42 d after flowering). (**A**) The AS patterns of genes in yellow- and black-seeded *B. napus* in five developmental stages of seed development (14, 21, 28, 35 and 42 d after flowering). The first row represents data from black-seeded samples, and the second from yellow-seeded samples. (**B**) Genes exhibiting AS at five developmental stages (14, 21, 28, 35 and 42 d after flowering). The pentagon at the center represents the 830 overlapping differentially spliced genes at the five specific time points. (**C**) Visualization of 830 DAS (differentially alternative splicing) genes identified using the enriched Gene Ontology (GO) biological process.

**Figure 2 plants-09-00977-f002:**
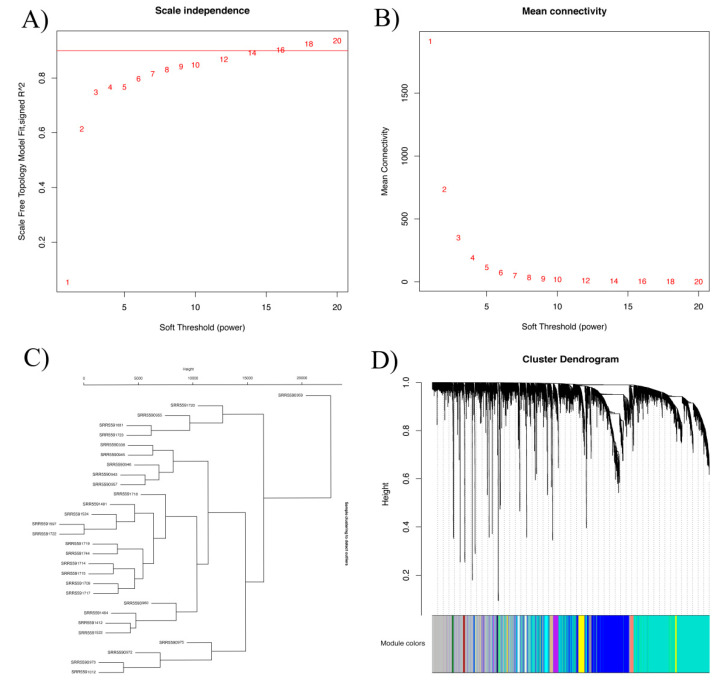
Module identification by weighted gene co-expression network analysis (WGCNA). (**A**,**B**) Network topology of the soft threshold with scale independence and mean connectivity. The approximate scale-free topology can be acquired at a soft threshold of 5. (**C**) Dendrogram of consensus module eigengenes obtained by WGCNA on the consensus correlation. (**D**) Gene dendrogram obtained by clustering the dissimilarity, based on consensus topological overlap with the corresponding module colors, indicated by the color row. Each colored row represents a color-coded module that contains a group of highly connected genes. A total of 23 modules were identified. The genes in the gray module are not significantly related, and the genes were not classified into any module.

**Figure 3 plants-09-00977-f003:**
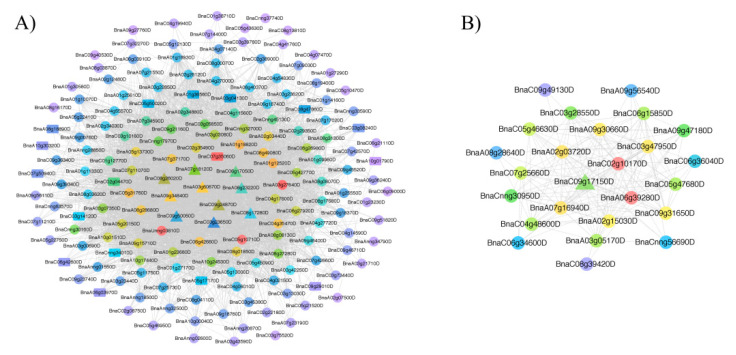
Visualization of two vital co-expression networks identified using Cytoscape. (**A**) the black module and (**B**) dark grey module. The candidate DAS genes are indicated by triangles, the intersection of genes in two vital functional modules and 830 overlapping genes are indicated by rectangles, whereas the remaining genes are indicated by circles.

**Table 1 plants-09-00977-t001:** Candidate DAS genes related to seed coat color in yellow- and black-seeded *B. napus*.

Gene	Homologous Genes	Annotated	Module	Stage
BnaA03g60670D	AT1G61720	BANYULS (BAN)	black	28 d
BnaC09g50050D	AT5G05270	Chalcone-flavanone isomerase family protein (CHI)	black	28 d
BnaC09g17150D	AT5G42800	dihydroflavonol 4-reductase (DFR)	dark grey	28 d, 35 d, 42 d
BnaA08g23220D	AT1G17260	autoinhibited H(+)-ATPase isoform 10 (AHA10)	black	14 d, 21 d, 28 d, 42 d
BnaC08g17280D	AT1G17260	autoinhibited H(+)-ATPase isoform 10 (AHA10)	black	21 d, 28 d
BnaAnng39120D	AT4G09960	SEEDSTICK (STK)	not included	14 d, 21 d, 28 d, 35 d, 42 d
BnaCnng46740D	AT4G09960	SEEDSTICK (STK)	dark red	42 d
BnaC08g21130D	AT3G49430	SER/ARG-rich protein 34A (SRp34a)	not included	14 d, 21 d, 28 d, 35 d, 42 d
BnaC01g26160D	AT3G49430	SER/ARG-rich protein 34A (SRp34a)	not included	14 d, 21 d, 28 d, 35 d, 42 d
BnaA06g15930D	AT3G49430	SER/ARG-rich protein 34A (SRp34a)	turquoise	14 d, 21 d, 28 d, 35 d, 42 d
BnaA09g34840D	AT3G55120	TRANSPARENT TESTA 5 (TT5)	black	28 d, 42 d
BnaC08g26020D	AT3G55120	TRANSPARENT TESTA 5 (TT5)	black	28 d, 35 d, 42 d
BnaC09g24870D	AT4G09820	TRANSPARENT TESTA 8 (TT8)	black	21 d, 28 d, 35 d, 42 d
BnaC06g17050D	AT3G59030	TRANSPARENT TESTA 12 (TT12)	black	21 d, 28 d, 35 d, 42 d
BnaA07g18120D	AT3G59030	TRANSPARENT TESTA 12 (TT12)	black	42 d
BnaAnng08030D	AT5G48100	TRANSPARENT TESTA 10 (TT10)	not included	21 d, 28 d, 35 d, 42 d
BnaA06g30430D	AT5G48100	TRANSPARENT TESTA 10 (TT10)	blue	21 d, 28 d, 35 d, 42 d
BnaC02g38340D	AT5G48100	TRANSPARENT TESTA 10 (TT10)	turquoise	28 d, 35 d, 42 d
BnaAnng30140D	AT5G23260	TRANSPARENT TESTA16 (TT16)	turquoise	14 d
BnaC09g04950D	AT5G23260	TRANSPARENT TESTA16 (TT16)	turquoise	42 d
BnaC04g41260D	AT5G03495	RNA-binding family protein	not included	14 d, 21 d, 28 d, 35 d, 42 d
BnaC07g01360D	AT5G03495	RNA-binding family protein	turquoise	14 d, 21 d, 28 d, 35 d, 42 d
BnaC03g02460D	AT5G03480	RNA-binding family protein	turquoise	14 d, 21 d, 28 d, 35 d, 42 d
BnaAnng15850D	AT5G03480	RNA-binding family protein	blue	14 d, 21 d, 28 d, 35 d, 42 d

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
