# Peer review of "Differences in Alternative Splicing between Yellow and Black-Seeded Rapeseed"

_plants, 2020, doi:10.3390/plants9080977_

Round 1

Reviewer 1 Report

The submitted study presents an alternative splicing comparison between yellow-seeded and black-seeded rapeseed. The concept of the submitted study is very interesting and it can be characterized as innovative because it deals with a vital post-transcriptional regulatory process, which contribute to plant cell differentiation and organ development. The number of references discussed in the "Introduction" are in a satisfactory level and also the main innovative characteristics of the present study against existing literature are well pointed out. The paper manuscript is of high caliber and it is worth to be published in the highly reputable journal "Plants" with only minor revisions. The minor issues that require attention are the following:

  1. According to my humble opinion the "Materials and Methods" section should be placed before the "Results" and the "Discussion". "Materials and Methods" concern the main body of the paper and afterwards the "Results and Discussion" should be placed.
  2. There is no "Conclusions" section. It is of utmost importance for the quality and the integrity of the article to add after the "Results and Discussion" section the "Conclusions" section, which should contain the main conclusions of the submitted study in bulleted form. 

Author Response

Dear Reviewers,

Thank you for your kind suggestions and comments. We sincerely appreciate your valuable comments, which not only helped us improve our manuscript, but also provide some good ideas for future research. We have studied your comments carefully and have made the required corrections. We hope that the revised version of our manuscript will meet with your approval. The main corrections and responses to your comments are listed below.

Best regards,

Jiana Li

List of responses:

The submitted study presents an alternative splicing comparison between yellow-seeded and black-seeded rapeseed. The concept of the submitted study is very interesting and it can be characterized as innovative because it deals with a vital post-transcriptional regulatory process, which contribute to plant cell differentiation and organ development. The number of references discussed in the "Introduction" are in a satisfactory level and also the main innovative characteristics of the present study against existing literature are well pointed out. The paper manuscript is of high caliber and it is worth to be published in the highly reputable journal "Plants" with only minor revisions. The minor issues that require attention are the following:

1. According to my humble opinion the "Materials and Methods" section should be placed before the "Results" and the "Discussion". "Materials and Methods" concern the main body of the paper and afterwards the "Results and Discussion" should be placed.

Response: Thank you for your suggestions. However, according to the Plants Microsoft Word template file, the "Materials and Methods" section should be placed after the "Results" and the "Discussion".

2. There is no "Conclusions" section. It is of utmost importance for the quality and the integrity of the article to add after the "Results and Discussion" section the "Conclusions" section, which should contain the main conclusions of the submitted study in bulleted form.

Response: Thanks a lot. The following sentences have been added in the "Conclusions" section after the "Results and Discussion" section. “In the study, we identified candidate alternative splicing genes between yellow-seeded and black-seeded oilseed rape during the post-transcriptional regulatory process. The function of candidate genes need to be verified by molecular biology experiments in the future. This study will help us to understand the formation mechanism of yellow-seeded B. napus.”

In the end, we are extremely grateful for your helpful suggestions. Thanks!

Reviewer 2 Report

This article describes differential splicing between yellow- and black-seeded rapeseed (Brassica napus) at seed development. The authors performed bioinformatics analyses of RNA-seq data of developing seeds of B. napus, and demonstrate that several key genes involved in flavonoid biosynthesis and SR protein genes show differential splicing between yellow and black seeds. The data provide some useful information about splicing variants and novel isoforms of B. napus genes related to seed coat color. However, there are several concerns in the current manuscript as follows. 

Major points

1) The authors used previously published RNA-seq data of B. napus for this study. Although the details of the samples of RNA-seq have been documented in a previous publication, brief description of the samples (ex. cultivars) should be presented.

2) According to Figure S1, there are large variations in FPKM values among the three replicates of the samples. For example, Y1d, B42d, and Y42d show variations of two orders of magnitude. What do the authors explain for these large differences. I wonder whether some samples used for RNA-seq might have problems which cause large variation in FPKM. If so, it would be better to exclude such samples from the analyses.

3) The authors mention that “AS mainly occurred in the early stage of seed development” in L20-21 in abstract and L122-123. What is the basis of this notion? The number of stage-specific DAS genes peaked at 28 days after flowering (L115). Therefore, it seems that AS mainly occurred not in the early but the middle stage of seed development.

4) The authors listed several key genes that showed differential splicing in Table 1. But I can’t understand the reason why some genes mentioned in the text are not listed in the table. Those genes are three DNA/RNA-binding protein genes (L111-112) and RNA-binding family protein gene (L180-181). And the title of the table is not appropriate, which I would suggest something like “Candidate DAS genes related to seed coat color in yellow- and black-seeded B. napus”. In addition, “unknown” in the “Module” column is not possible since the authors must be able to identify which gene belongs to which modules. Therefore, this word should be replaced to some module name or “not included” (in any modules).

5) This study revealed the global differences in splicing during seed development, but the discussion about the significance of those differences is scarce in the current manuscripts. Are there any changes in coding sequences or expression levels by alternative splicing in the DAS genes listed in Table 1? I think the authors can demonstrate some examples and discuss the influence of alternative splicing of those genes on seed coat color.

6) There are several problems concerning supplementary figures and tables. Since some figures and tables have no legend, I don’t understand what are shown in them. For example, what do the rectangles, arrows, and red and orange colors represent in Figure S1? What do “u” and “all” indicate in Table S1? It seems that Table S1 and S2 show exactly same data and are redundant. Also, I think Table S3 is not necessary because it is not mentioned in the main text.

Minor points

7) L20: Since “intron retention” appears only once in Abstract, I think there is no need to abbreviate this phrase.

8) L22: I would suggest that “composed of genes” should be deleted or changed to “composed of differentially spliced genes”.

9) L23: “models” should be changed to “modules”.

10) L24: What does “DAS” represent? Please do not use abbreviations without definition.

11) L31: I think “Brassica napus” should be included in Keywords.

12) L36: Ref 3 is a review paper published in 1986, which is not appropriate to cite. Original work published in 1981 should be cited as Ref 3.

13) L38: “more” is not necessary.

14) L59-60: This sentence mentions alternative splicing in plants, However, a study in human is cited here (Ref 18), which should be replaced.

15) L77-78: I think this sentence is grammatically incorrect. These genes are involved in “what function”?

16) L97-98: There are several abbreviations defined in this sentence, but they appear in L42-44 for the first time.

17) L100: “flowering” should be changed to “seed development”.

18) L105: Figure 1B must be cited as well as TableS4.

19) L106-107: I can’t find the GO category “nitrogen compound transport” in Figure 1C, and “potassium ion transport” is displayed in the figure instead.

20) L110-111: Ref 26 is not a study of Arabidopsis but rice.

21) L129-130: It would be better to write “at five developmental stages of seed development (14, 21, 28, 35, and 42 days after flowering) instead of “on five specific days after flowering (14 d, 21 d, 28 d, 35 d, 42 d)” to. I would suggest changing the phrases in L131 and L133 accordingly.

22) L134: The central area of 830 overlapping genes is not a “dark blue rectangle”. It must be called “pentagon at the center”.

23) L140: “to explore genes” is not necessary.

24) L213: I think ”bZIZIP6” must be “bZIP6”.

25) There are several problems in References. Some Refs are not written in the journal format. The journal name of Ref 9 is written in Chinese, which should be in English. Ref 26 and 44 are a same paper. There is not enough information in the list to identify Ref 51 and 52. These are only some examples, and I’d like the authors to check and correct the reference list carefully.

Round 2

Reviewer 2 Report

In the revised manuscript, the authors responded to my comments properly. I found that the revised manuscript has been quite improved, and I appreciate their effort to complete the revision in a short period of time.